# No relation between Body Mass Index and neurocognitive recovery in abstinent alcohol dependent patients?

Jeroen Staudt[1,2]*, Yvonne Rensen[2,3], Hein De Haan[1,4], Boukje Dijkstra[5,6], Jos Egger[2,3,7]

**1** Tactus Institute for Addiction Care, Deventer/Zutphen, The Netherlands, **2** Donders Institute for Brain, Cognition and Behaviour, Radboud University, Nijmegen, The Netherlands, **3** Centre of Excellence for Korsakoff and Alcohol Related Cognitive Disorders, Vincent van Gogh, Venray, The Netherlands, **4** Forensic Psychiatry Department de Boog, GGNet, Warnsveld, The Netherlands, **5** Nijmegen Institute for Scientist Practitioners in Addiction (NISPA), Radboud University, Nijmegen, The Netherlands, **6** Novadic-Kentron, Vught, The Netherlands, **7** Centre of Excellence for Neuropsychiatry, Vincent van Gogh, Venray, The Netherlands

* jeroen.staudt@donders.ru.nl

## Abstract

### Background

Neurocognitive deficits in patients with Alcohol Use Disorder (AUD) may partially arise as a result from nutritional deficiencies. Previous studies have shown associations between nutritional deficiencies and impaired neurocognitive function, but the results are inconclusive. Difficulties with operationalization may play a role in this. This study examined whether nutritional deficiencies, reflected by Body Mass Index (BMI) at admission, predict neurocognitive functioning after six weeks of abstinence and recovery of neurocognitive functions. This was also examined for different groups of BMI (very low, normal and very high).

### Methods

Data was derived from 86 patients who were diagnosed with AUD following DSM-5 criteria and referred for clinical detoxification and neuropsychological examination. At admission, BMI, demographic information, and information regarding alcohol use were derived from the medical record and a clinical interview. During the second and sixth week of admission patients underwent neuropsychological assessment using well normed and validated instruments. Both longitudinal data and cross-sectional data from the sixth week were used to examine neurocognitive recovery.

### Results

BMI as a continuous and categorical measure predicted the score on two tasks measuring speed and visuomotor performance after six weeks of abstinence. No further

**Data availability statement:** The data that support the findings of this study contain potentially identifying and/or sensitive patient information and hence are only available upon reasonable request to the scientific committee of Tactus addiction treatment (weco@tactus.nl). The provision of data will be considered by the Tactus Local Scientific Research Committee and the authors involved in this study. The sharing of patient data is subject to Dutch and European legal and ethical regulations.

**Funding:** The author(s) received no specific funding for this work.

**Competing interests:** The authors have declared that no competing interests exist.

relation between BMI with neurocognitive function or recovery of neurocognitive functions was found.

## Conclusions

This study found tentative support for BMI to predict performance on speed and visuomotor functioning in AUD-patients. These results may partly support an inverted U-shape, in which a very low BMI is negatively related to the outcome. Given the influence of nutritional deficiencies on the development of neurocognitive disorders, there is a need for further research that takes into account a possible non-linear relationship between BMI and neurocognitive functions, using additional physical measures, to identify (past) nutritional deficiencies.

## Introduction

Chronic, excessive alcohol use affects the health of a variety of organ systems among which the gastrointestinal tract, resulting in problems with digestion, absorption and metabolic processes [1,2]. This particularly puts people with alcohol use disorder (AUD), who often already have malnutrition resulting from a poor diet, at risk for even more severe nutritional deficiencies [1,3,4]. It is not uncommon that patients with AUD have neurocognitive deficits as a consequence of the direct neurotoxic effects of alcohol and the indirect effects from malnutrition [5,6]. Moreover, nutritional deficiencies might contribute to the development of neuropathological disorders (i.a., Korsakoff's syndrome, Marchiafava Bignami, Pellagra, Wernicke Encephalopathy(WE), [5,7–9].

The relationship between nutritional deficiencies and cognitive functioning has long been of scientific interest. Rosenthal and Goodwin [10] pointed out the need to study this relationship. Yet the number of studies available to date are scarce [7,11–15] and the results are inconclusive. Nutritional deficiencies in these studies is operationalized in different ways. Blood values are used to examine the relation between thiamine diphosphate (TDP) and neurocognitive functioning. TDP, as a key factor driving cognitive impairments [16], was significantly associated with neurocognitive functioning in four studies [7,11,14,15], but not in two other studies [12,13]. Another operationalization of nutritional deficiencies is the number of 'missed meal days'. Two studies, using group classifications according to subclinical forms of WE, found significant negative associations between '30 missed meal days' and neurocognitive functions (a.o. IQ, memory, visuoconstruction and processing speed) [7,14]. Another study did not find a significant relationship between 30 missed meal days and neurocognitive functioning (executive function, visuospatial abilities, verbal episodic memory) [15]. Finally, two studies used Body Mass Index (BMI, measured by weight in kilograms divided by the height in meters (squared)) to operationalize nutritional status, and the results are inconclusive. One study reported a significant relationship between BMI measured in the first week of admission and performance on a neurocognitive screener after 10 days of abstinence [12]. The other study

reported no significant relationship between BMI at admission and a cognitive screener at multiple assessment points [11]. The few studies, using various operationalizations of (actual and past) nutritional deficiencies, show inconclusive findings and illustrate a need for further clarification.

Several factors need to be taken into account, when examining the relation between nutritional deficiencies and cognitive functioning. First, there is no gold standard to operationalize nutritional deficiencies, and the construct is inexact and difficult to assess [17]. The (understudied) construct of BMI may be considered an appropriate measure reflecting nutritional deficiencies in patients with AUD. BMI has been significantly related to nutritional status (by means of the Subjective Global Assessment) in patients with [18] and without AUD [19,20], is associated with alcohol related complications such as liver disease [1], and is part of standard physical examination in addiction healthcare (according to the guidelines from the National Institute for Health and Care Excellence [NICE], Alcohol-use disorders: diagnosis and management) [21,22]. Yet, only two studies investigated the relation between BMI and neurocognitive functioning, by measuring BMI at the start of a clinical admission and assessments of neurocognitive functioning (MoCA) during their stay [11,12]. In addition, the relation between BMI and neurocognitive functioning may not be linear, since excessive alcohol use seems to increase BMI to a certain point, after which alcohol consumption accelerates metabolism and decreases fat mass, causing an inverse relation with BMI [17,23]. Several mechanisms may underlie this reversed U-shape, e.g., an increased microsomal detoxification or reduced lipogenesis (see [17]). Though an exact tipping point is unknown, this curve is also being studied in other areas of healthcare [24] and may be used to model the relation between malnutrition and neurocognitive functioning.

Second, there is noticeable diversity in the duration of abstinence from alcohol among studies, while it is known that duration of abstinence may affect neurocognitive performance [25,26]. In the studies by Fama and colleagues [7] and Pitel and colleagues [14], the variation in abstinence duration was considerable (i.a., 16 weeks, SD = 1–746 days), while in other studies [11–13,15], the abstinence duration was relatively short (< 31 days abstinent, see [25]). Assessing neurocognitive performance after six weeks of abstinence is recommended, because of cognitive recovery after alcohol detoxification [27].

Third, next to abstinence, a number of other factors may affect neurocognitive functioning [28,29]. Tobacco smoking should in any case be considered given the strong evidence for a negative relationship with neurocognitive recovery [28,30], but also other factors such as gender and education level.

Finally, all studies examined relations between (aspects of) nutritional deficiencies and measures of neurocognitive functioning at one assessment point in time. By studying the relationship between nutritional deficiencies and *differences* between multiple assessments of neurocognitive function, insight might be gained into the extent to which nutritional deficiencies may affect recovery of neurocognitive functioning [28,31].

In summary, studies on the relation between nutritional deficiencies and neurocognitive functioning in patients with AUD are scarce, face methodological challenges and thus far present inconclusive findings. BMI, in this regard, is considered an appropriate, non-invasive, measure reflecting nutritional status, and a feasible outcome measure since it is part of standard physical examination in addiction care. Given the limited amount of studies on the relation between nutritional deficiencies and neurocognitive functioning among abstinent patients with AUD, and the various ways in which nutritional deficiencies are operationalized, this study explores the usability of BMI at admission, as a predictor of neurocognitive functioning and/or recovery of neurocognitive functions during abstinence.

The aim of this explorative study is to examine the relation between BMI and neurocognitive functioning in patients with AUD. First, we examine if a continuous BMI value predicts neurocognitive functioning after six weeks of abstinence in patients with AUD. Given the possibility of an inverted U-shaped relation we also examine whether categories of BMI (very low, normal, very high) predict neurocognitive function. Second, we examine the relation between BMI (continuous and categorical) and *recovery* of neurocognitive functioning using multiple assessments. In order to accurately examine these relationships we use an associative modelling approach [32].

## Materials and methods

### Design and participants

For this retrospective cohort study, anonymized data collected from standard care procedures at specialized clinical addiction healthcare facilities in the eastern and southern part of the Netherlands were used (Tactus addiction healthcare and Vincent van Gogh Centre of Excellence for Korsakoff and Alcohol-Related Cognitive Impairments (VvG). Both longitudinal and cross-sectional data were collected between January 2013 and December 2020, and accessed for analysis in April and May 2024. The first author received anonymized data from VvG, which could not be traced back to individual participants. Data collected at Tactus was anonymized by means of an encryption file used by the first author. The authors had no access to information that could identify individual participants. The healthcare facilities participate in the NISPA collective (Nijmegen Institute for Scientist Practitioners in Addiction healthcare) and conduct neuropsychological research in similar ways. Participants who met criteria for a DSM-5 AUD disorder [33] were included for analyses. Data from participants were excluded in case of pregnancy, severe somatic or neurological disorders (e.g., lupus, congenital epilepsy, diagnosis of acquired brain injury), acute psychiatric disorder (psychosis, suicidal tendencies), or insufficient Dutch language skills. The research was performed in accordance with the Declaration of Helsinki and executed following the guidelines of the Tactus Local Scientific Research Committee and the Vincent van Gogh Institutional Review Board. All participants gave written informed consent and patients classified with AUD were included.

### Procedure

Patients were referred to the aforementioned facilities for inpatient alcohol detoxification and cognitive assessment and they received extensive neuropsychological examination. Depending on the progress during admission, the length of their stay varied from six weeks up to three months. At admission information on weight in kilograms and length in centimetres was collected, as well as demographic information (gender, age, level of education), information about the use of alcohol (AUD, number of years and age of onset of daily alcohol use) and tobacco were collected.

As part of routine care, patients firstly underwent screening of neuropsychological functions by means of the MoCA [34] in the second and sixth week of admission. Secondly, after sixth weeks patients also underwent neuropsychological assessment in which frequently affected domains were investigated. Abstinence from alcohol and other substances during admission was verified by urinalyses, twice per week. Only data from abstinent participants is used for the purpose of this research. All neuropsychological assessments were performed by a trained psychologist.

### Measurements

**Clinical interview for demographic and drinking related information.** Demographic information (gender, education level and age) and quantities of previous alcohol use (e.g., 'the number of years of daily alcohol use', 'the age of onset of daily alcohol use') were registered via a clinical interview based on the Measurements in the Addictions for Triage and Evaluation (MATE) [35,36]. The level of education was classified according to Verhage's model [37], and classified as 'Low' (levels 1 and 2), 'Average' (levels 3, 4 and 5) and 'High' (levels 6 and 7). Substance use disorder(s) were classified following the DSM-5 criteria [33].

**Body Mass Index (BMI).** BMI was derived from the medical report at admission and calculated by dividing the weight in kilograms by length in metres squared. To examine group differences, the BMI variable was translated into three categories (very low, normal, very high) following CDC guidelines [38] and previous studies [24]. A BMI value between 18.5 and 24.9 is considered 'normal', BMI values lower than 18.5, as 'very low' and higher than 24.9 as 'very high'.

**Montreal Cognitive Assessment (MoCA).** The MoCA is a concise screening tool for measuring cognitive functioning using 8 domains of neuropsychological functioning (executive functioning, visuospatial skills, concentration and work pace, language, memory and orientation) [34]. The instrument is available in 3 parallel versions (7.1, 7.2 and 7.3) and

a newer 8.1 version. Standardly version 7.1, or 8.1 with patients admitted more recently, was used during the first assessment. During the second assessment the 7.2 version was used to minimize learning effects. Each assessment resulted in a total score which was used for the purpose of this research. A score < 26 out of 30 is indicative of cognitive disorder, representing good sensitivity and specificity [34]. The instrument is translated in Dutch, shows a good test-retest reliability and is well normed [39,40].

**d2 Test of Attention.** The d2 Test of Attention is a paper and pencil cancellation task, measuring selective and sustained attention [41]. The scaled T-scores (T = ($\bar{x}$ - $\mu$)/ (s/ $\sqrt{n}$)) for concentration performance (total number of correctly crossed out items, minus the errors of commission), processing speed (total number of characters processed) and error percentage (accuracy rate) were used as outcome variables. Internal reliability is high for all outcome measures used (Concentration Performance 0.97, Processed Targets 0.97 and Error Percentage 0.92). Test-retest correlations are high to very high [41]. Normgroups consist of people between 18 and 80 years.

**D-KEFS trail making test.** This is a paper and pencil test from the Delis–Kaplan Executive Function System (TMT D-KEFS) and measures cognitive set shifting, visual scanning, visual motor coordination, alphabetical knowledge and numerical knowledge [42,43]. The scaled scores (Mean = 10, Sd = 3) on visual scanning (card 1), digit sequence (card 2), letter sequencing (card 3), digit-letter sequencing (card 4) and motor speed (card 5) were used as outcome variables. Internal consistency for the age groups used was medium to high [43,44]. Test-retest reliability lies predominantly within the medium range [43,44]. D-KEFS tests has been subject of multiple validity studies in various populations, demonstrating sensitivity to the assessment of executive (dis)functioning [44]. Normgroups consist of people between 8 till 89 years.

## Sample size calculation

Sample size was determined using G*power [45]. We calculated an approximation of the sample size for an *F* test (multiple linear regression). Based on previous studies [12] we expected a small to medium effect size ($f^2$ = 0.20). Alpha was set at 0.05 and we accepted a power of 0.80. We estimated the maximum number of predictors to be used at five. For the analysis of both our cross-sectional and longitudinal data a sample size of 70 participants was calculated in order to detect an effect using multiple linear regression. In previous studies 80–96 participants [7,12] were included in order to determine whether significant associations could be found on comparable neurocognitive tasks.

## Statistical analyses

Baseline characteristics were described using the appropriate measures for descriptive statistics. Normality was judged by visual inspection of the data. In case of doubt, skewness and kurtosis were calculated to Z-scores [see for example [46]] where a score between −2 and +2 was accepted for normality [47]. Neurocognitive functioning, as our dependent variable, is reflected by the scores on MoCA 1, MoCA 2, D-KEFS TMT and d2 task. Recovery of neurocognitive functioning is reflected by using the scores on MoCA 1 and MoCA 2 as repeated measures. All significant relations (p < 0.15) between BMI and neurocognitive functioning, or BMI and neurocognitive recovery, represent a model and were examined for confounding by determinants (gender, age, education level, severity of alcohol use disorder, tobacco use disorder (yes/no), years of daily alcohol use, age of onset of daily alcohol use and neurological injury (yes/no)) using the following steps. A variable was considered a candidate confounder if it was associated with both neurocognitive functioning and BMI (Chi-square, parametric or non-parametric tests). A more tolerant p-value (p < 0.15) is used to minimalize type II error. Subsequently, following association modelling [32], only one confounder at the time was added to the existing significant model between BMI and the dependent variable. Confounding is considered when the regression coefficient changes by 10% [32]. Associations were checked for multicollinearity using the Variable Inflation Factors index.

In order to examine the relation between BMI (as a continuous or a categorical measurement) and neurocognitive functioning, we used multiple linear regression analysis (Enter method, p < 0.15). BMI categories were examined using dummy

variables, with the normal BMI category as the reference group. Since missing variables were missing at random, these were listwise removed from analyses. For the second research question, examining the relation between BMI and *recovery* of neurocognitive functioning, we used linear mixed models. Two models were calculated for both a continuous value of BMI and a categorical value of BMI (very low, normal, very high) as fixed factors, with the scores on both MoCA tasks (repeated measures) as the dependent variable. The effect of time, and the interaction between BMI and time, as fixed factors, were also examined ($p < 0.15$). The intercept is considered as a random factor.

Given the explorative nature of this study, no correction of type I error was carried out. Statistical analysis was carried out using the Statistical Package for Social Sciences software, SPSS, IBM (version 29).

## Results

A total of 86 patients were included in this study. Data on almost all demographic variables were available, with exception of 'years of daily alcohol use' (N = 59), 'age of onset of daily alcohol use' (N = 63) and 'neurological injury' (N = 77). Due to variations in the administered tests during the seven years of data collection, 23 participants did not complete the first MoCA, 26 did not complete the second MoCA, 26 participants did not complete the d2, 13 participants did not complete the TMT, and 14 participants did not complete the TMT card 4.

### Description of the participants and outcome measures

The sample consisted largely of male patients (69%), see Table 1. On average patients were 52.1 years old and had an average level of education (Verhage levels 3–5, 9–13 years of education). A substantial part (54%) of the patients were

**Table 1. Demographic and AUD related variables of patients with AUD (n = 86).**

| Variable | | N |
|---|---|---|
| Gender, n (%) | | 86 |
| Men | 59 (69%) | |
| Women | 27 (31%) | |
| Age (years)[mean (SD)] | 52.1 (14.92) | 86 |
| Education level, n (%) | | 86 |
| Low (Verhage 1–3) | 21 (25%) | |
| Average (Verhage 4–5) | 48 (56%) | |
| High (Verhage 6–7) 1 | 16 (19%) | |
| Body Mass Index [mean (SD)] | 24.27 (6.60) | 86 |
| Body Mass Index groups, n (%) | | 86 |
| Very low BMI (< 18.5) | 9 (11%) | |
| Normal BMI (18.5–24.9) | 48 (46%) | |
| Very high BMI (> 25.0) | 29 (43%) | |
| Severity of AUD n (%) | | 86 |
| Light | 1 (1%) | |
| Moderate | 7 (8%) | |
| Severe | 78 (91%) | |
| Tobacco use disorder, n (%) | 62 (77%) | 81 |
| Years of daily alcohol use [mean (SD)] | 25.11 (18.61) | 59 |
| Age of onset of daily alcohol use [mean (SD)] | 33.22 (16.34) | 63 |
| Neurological injury, n (%) | 15 (19%) | 77 |

**Notes:** Education level according to Verhage (1964) classification, (Severity of) AUD and Tobacco Use Disorder according to DSM-5 criteria.

Abbreviations: SD (Standard Deviation), AUD (Alcohol Use Disorder).

classified as having a normal or very high BMI. However, only a few patients were classified as having a very low BMI (N = 9). During cognitive screening after two weeks of admission, 56% of the participants scored below the cut-off and after six weeks of admission, 65% scored below the cut-off. With regard to alcohol use, most patients were classified with severe AUD (91%) and used tobacco (77%). The average age at which daily alcohol use began was 33.2 years, with patients using alcohol daily for an average of 25.1 years.

Results regarding the outcome measures are presented in Table 2 and Table 3.

Average results of both MoCA screenings were below the cut-off score of 26.

**Table 2. Neurocognitive performance of patients with AUD and known BMI (n = 86).**

| Test | Mean score Mean (SD) | Range Min – Max | N |
|---|---|---|---|
| MoCA 1 | 23.52 (5.14) | 7–30 | 63 |
| MoCA 2 | 23.10 (4.16) | 6–31 | 60 |
| d2 N signs | 43.38 (9.13) | 30–69 | 60 |
| d2 N faults | 50.27 (10.11) | 30–70 | 60 |
| d2 CP | 42.80 (8.05) | 30–61 | 60 |
| d2 TV | 49.73 (10.65) | 30–70 | 60 |
| TMT card 1 | 9.66 (3.28) | 1–15 | 73 |
| TMT card2 | 8.16 (3.55) | 1–15 | 73 |
| TMT card 3 | 8.01 (3.88) | 1–15 | 73 |
| TMT card 4 | 7.42 (3.76) | 1–16 | 72 |
| TMT card 5 | 10.55 (2.30) | 2–15 | 73 |

Abbreviations: SD, Standard Deviation, AUD, Alcohol Use Disorder, MoCA, Montreal Cognitive Assessment, d2 = d2 Test of Attention, with N signs = total number of signs processed, N faults = total number of faults made, CP = composite variable representing concentration, TV = Tempo Variation, TMT = Trail Making Test consisting of 5 subtasks (card 1–5).

**Table 3. Neurocognitive performance per BMI category.**

| Test | Normal BMI (n = 48) | | | Very low BMI (n = 9) | | | Very high BMI (n = 29) | | | P1 | P2 |
|---|---|---|---|---|---|---|---|---|---|---|---|
| | Mean score (SD) | Range Min - Max | N | Mean score (SD) | Range Min - Max | N | Mean score (SD) | Range Min - Max | N | | |
| MoCA 1 | 23.71 (5.82) | 7–30 | 34 | 21.75 (3.96) | 16–27 | 8 | 23.90 (4.37) | 11–28 | 21 | | |
| MoCA 2 | 23.66 (3.29) | 17–28 | 35 | 23.33 (3.83) | 17–27 | 6 | 22.00 (5.51) | 6–31 | 19 | | |
| d2 N signs | 43.23 (8.57) | 30–60 | 35 | 46.38 (12.32) | 30–69 | 8 | 42.29 (8.89) | 30–58 | 17 | | |
| d2 N faults | 49.80 (10.15) | 30–70 | 35 | 49.38 (12.72) | 30–61 | 8 | 51.65 (9.17) | 31–64 | 17 | | |
| d2 CP | 42.51 (8.10) | 30–61 | 35 | 42.25 (8.58) | 30–51 | 8 | 43.65 (8.15) | 30–57 | 17 | | |
| d2 TV | 50.40 (11.12) | 30–70 | 35 | 45.88 (12.54) | 30–67 | 8 | 50.18 (8.85) | 39–63 | 17 | | |
| TMT card 1 | 9.28 (3.10) | 1–14 | 40 | 9.11 (3.86) | 1–13 | 9 | 10.50 (3.34) | 1–15 | 24 | | |
| TMT card2 | 7.65 (3.40) | 1–15 | 40 | 7.11 (4.40) | 1–12 | 9 | 9.42 (3.46) | 2–15 | 24 | | * |
| TMT card 3 | 7.42 (3.79) | 1–15 | 40 | 9.00 (3.91) | 6–11 | 9 | 8.62 (4.02) | 1–15 | 24 | | |
| TMT card 4 | 6.64 (3.53) | 1–12 | 39 | 9.00 (1.66) | 2–13 | 9 | 8.08 (4.44) | 1–16 | 24 | * | |
| TMT card 5 | 10.28 (2.33) | 3–14 | 40 | 9.67 (3.20) | 9–12 | 9 | 11.33 (1.66) | 9–15 | 24 | | |

Notes: P1 = significant difference (p < 0.5) between normal BMI and very low BMI, P2 = significant difference (p < 0.5) between normal BMI and very high BMI

Abbreviations: SD, Standard Deviation, AUD, Alcohol Use Disorder, MoCA, Montreal Cognitive Assessment, d2 = d2 Test of Attention, with N signs = total number of signs processed, N faults = total number of faults made, CP = composite variable representing concentration, TV = Tempo Variation, TMT = Trail Making Test consisting of 5 subtasks (card 1–5)

### Continuous BMI and neurocognitive functioning

BMI significantly predicted the score on TMT card 5 ($R^2$ = .055, F (1, 71) = 4.169, p = .045), showing that an increase of BMI predicted an increase on the score on TMT card 5, by approximately .08 percentage point. Confounder analysis yielded 'Years of AUD' as a candidate confounder in this relation, as this factor correlated with both BMI (p = .094) and D-KEFS TMT 5 (p = .122). Adding this factor to the regression equation yielded a non-significant model ($R^2$ = .097, F (1, 50) = 2.690, p = .078). BMI did not predict other measures of neurocognitive functioning.

### Categorical BMI and neurocognitive function

A regression model predicting performance on TMT card 2 by means of categories of BMI yielded a significant result ($R^2$ = 0.064, F (2, 70) = 2.397, p = 0.098). The contribution of the very high BMI category was significant (β = 0.235, p = 0.054), the very low BMI category did not contribute significantly (β = −0.050, p = 0.673). No other determinants could be identified as a candidate confounder in this relationship.

A regression model predicting performance on TMT card 5 by means of categories of BMI yielded a significant result ($R^2$ = 0.065, F (2, 70) = 2.427, p = 0.096). The contribution of the very high BMI category was significant (β = 0.217, p = 0.074), the very low BMI category did not contribute significantly (β = −0.087, p = 0.468).

Two candidate confounders were identified. Adding age to the equation yielded a significant model ($R^2$ = 0.106, F (3, 69) = 2.738, p = 0.050), with a positive effect of age on the dependent variable (β = 0.206, p = 0.078). The beta coefficient of the very low BMI category decreased > 10% from −0.087 to −0.075. Adding Age of Onset Of Daily Alcohol Use to the model did not yield a significant model ($R^2$ = 0.085, F (3, 53) = 1.631, p = 0.193).

### Continous BMI and neurocognitive recovery

Results from linear mixed model analysis yielded no significant effect of time (p = .713), BMI (p = .496) or time * BMI (p = .634) on repeated measures of both MoCA assessments

### Categorical BMI and neurocognitive recovery

Results from linear mixed model analysis yielded no significant effect of time (p = .452), BMI category (p = .689) or time * BMI category (p = .168) on repeated measures of both MoCA assessments.

## Discussion

In the current study we investigated if nutritional deficiencies, measured using BMI at admission, predicted speed, attention, and executive functioning and *recovery* of neurocognitive functioning in abstinent patients with AUD. Using an association modelling approach, we analysed the relation between BMI, neuropsychological functioning and recovery, and candidate confounders (gender, age, education level, severity of alcohol use disorder, tobacco use disorder, years of daily alcohol use, age of onset of daily alcohol use and neurological injury).

With regard to our first research question, we found that both the continuous, and the categorical BMI value, correlated with a task mainly tapping speed and visuomotor skills (TMT-5). The categorical BMI value also correlated with a comparable task tapping digit sequencing, next to speed and visuomotor skills (TMT-2). Visuospatial and motor impairments may endure after detoxification [25,48,49] and may be related to BMI at admission. Interestingly, the results showed differences in the contribution of the categories of BMI (very low, normal and very high) to the models. The category of very low BMI was negatively related to performance on both the TMT-2 and TMT-5 tasks, the category of a very high BMI on the other hand was positively related to the outcomes. Although only the contribution of the very high BMI group was significant, possibly due to the very small sample size of the very low BMI group (N = 9), these results may be indicative of a difference in performance between the groups with the very low group performing worse. This may partly support an

inverted U-shaped relation between BMI and speed -and visuomotor performance. The results furthermore show that the factor Age should be considered as a confounder in this relationship. With increasing age, performance on one task (TMT-5) increased. Since older adults show an increased chance of visual -and motor control deficits [50], this finding seems counterintuitive and might be attributed to the aforementioned skewed BMI data.

With regard to our second research question, the results did not support evidence for a relation between BMI at admission with recovery of neurocognitive function, as measured with two cognitive screening instruments during short (2 weeks) and mid-term (6 weeks) abstinence. Next to the small sample of participants with a very low BMI, this null finding may be related to the use of the MoCA as a cognitive screener to detect change in cognitive function over time. Though the MoCA is considered a suitable tool for detecting cognitive impairment in populations with substance use disorders [51], the total score is used as a cut-off and not as a continuous measure. The instrument is not as sensitive as conducting a full neuropsychological examination and may therefore not accurately capture subtle differences in neuropsychological functioning. Moreover, the abstinence duration and the time between the assessments is relatively short, while recovery of neurocognitive functions can take longer [25,49], thus also complicating the detection of recovery.

In the current study, we generally found no evidence for a relation between BMI and neurocognitive function, or recovery of neurocognitive functions, after six weeks of abstinence. Tentative evidence was found for a relation between BMI at admission and speed -and visuomotor performance. The findings may reflect a part of the so called inverted 'U-shape' in which participants with a very low BMI may perform substantially lower than participants with a normal or very high BMI. This part of the results add to the evidence of a study in which a continuous BMI measure at admission predicted MoCA scores after 10 days of abstinence [12]. The most of our (null) findings are however in line with findings of the study done by Ritz et al. [15], in which no relation was reported between current malnutrition (continuous BMI was one of the factors comprising the construct) with memory or executive functions.

Nutritional deficiencies play an important role in increasing the risk of developing neurocognitive disorders [5,14]. Although BMI is considered an appropriate measure reflecting nutritional deficiencies, BMI alone is not considered a valid predictor of neurocognition in patients with AUD for several reasons. The complex interplay of dietary habits with gastrointestinal illness, malabsorption and dysfunction of microbiota [2], may not always be reflected by deviating BMI values. For example, patients with AUD have presented with nutritional deficiencies while BMI values were within the normal range [1]. This raises the question whether other (non-invasive and easily available) measures should also be used that may be able to more accurately capture nutritional status in this population. Abdominal waist circumference or waist-to-hip ratio are also studied in this regard [52,53] and are known to predict underlying (alcohol related) liver disease more accurately than BMI [54]. Given the association known between liver disfunction, nutritional deficiencies and neurocognitive disfunction [5,15], future studies may consider incorporating these factors, and to repeatedly assess these measures over time to examine whether change in nutritional deficiencies parallels changes in cognitive functioning. However, it should be noted that these measurements may not be easy to obtain as they are not part of standard physical examination [21]. Although it is recognized that past nutritional deficiencies may play an important role in predicting neurocognitive impairments and -recovery, to date no reliable and valid measurements are available to showcase this notion [7,14]. Moreover, the current nutritional status does not necessarily correspond to past nutritional status. It is possible that (irreparable) cognitive damage has already occurred due to a previous nutritional deficiency, but the nutritional status of that time cannot be determined using current BMI or current blood work. Future studies should make effort to incorporate biological and behavioural measures of past, and present, factors indicative of nutritional deficiencies. With regard to BMI, it is recommended to add abdominal waist circumference or waist-to-hip ratio, in order to determine which measure, or combination of measures, best predicts the neurocognitive outcome measure. Finally, given the aforementioned tipping point, and our results, it should be considered that BMI is not linearly related to nutritional status and neurocognitive functioning. Distinguishing groups of patients by multiple measurements reflecting nutritional health therefore seems appropriate.

This study comes with the following strengths. The naturalistic character of this observational two centre study, using data from two different addiction health care centres is reflective of the population and increases the external validity of the findings. The research design, using both longitudinal and cross-sectional data across a span of six weeks of abstinence, allowed an investigation of the relation between BMI and recovery of neurocognitive function. Lastly, this is the first study known to us to explicitly investigate the relation between neurocognitive impairments and three groups of participants according to their BMI status. Next to strengths, this study also comes with limitations. The association between BMI and neurocognitive functioning was examined using instruments that predominantly tap speed, attention and executive functions. In addition to the clear and distinct effects from chronic alcohol use on these functions [25,55], it is known that cognitive impairments in patients with AUD are heterogenic in nature [56]. Future studies should therefore consider to investigate associations between nutritional deficiencies with other domains of neurocognitive functioning, e.g., memory and social cognition. In addition, the MoCA instrument used is a screening tool. Though it is ideally suited for multiple testing, given the parallel forms, it should be noted that the MoCA has less specificity than a full neuropsychological assessment [57,58]. Furthermore, this study did not include other measures of illicit drugs use, or nutritional deficiencies (e.g., whole blood thiamine, vitamins, missed meals), while this could have furthered our understanding of the relation between this complex construct and neurocognitive functioning. It may be worthwhile to consider the use of standardized instruments in order to assess nutritional status since this was not carried out in previous studies [59]. Lastly, the sample size was not sufficient for several analyses. First, the number of patients with a very low BMI value was small. Second, the amount of patients who completed both MoCA measures was smaller than necessary according to our power calculation (42 of 86). Although for all other analysis our sample size, used to perform our statistical analysis, was similar to the sample size used in other relevant studies [11–14], we ran multiple (12) models, risking type I error. Therefore the (limited) positive results cannot be excluded as false positives and need replication.

To conclude, this study found generally no support for a relation between BMI at admission and neurocognitive function, or recovery of neurocognitive functions, after six weeks of abstinence. The finding that participants with AUD and a very low BMI, but not a normal or very high BMI, perform worse on tasks assessing speed -and visuomotor functions after six weeks of abstinence may support part of the inverted U-shape relation between BMI and neurocognitive function. Future studies on the relation between nutritional deficiencies and neurocognitive functioning are needed, where attention should be paid to identifying nutritional deficiencies in different ways (biological and behavioral), distinguishing groups based on BMI categories given the possible non-linear relationship between BMI and neurocognitive functioning, the use of standardized instruments and taking into account a sufficient duration of abstinence.

## Acknowledgments

We would like to thank Dr. Serge Walvoort and Dr. Tim Kok for their constructive feedback during the drafting of this manuscript and all patients and professionals that participated in the data collection.

## Author contributions

**Conceptualization:** Jeroen Staudt, Yvonne Rensen, Hein De Haan, Boukje Dijkstra.

**Data curation:** Jeroen Staudt.

**Formal analysis:** Jeroen Staudt.

**Investigation:** Jeroen Staudt.

**Methodology:** Jeroen Staudt, Yvonne Rensen, Hein De Haan, Boukje Dijkstra.

**Project administration:** Jeroen Staudt.

**Supervision:** Yvonne Rensen, Hein De Haan, Boukje Dijkstra, Jos Egger.

**Validation:** Yvonne Rensen, Hein De Haan, Boukje Dijkstra, Jos Egger.

**Writing – original draft:** Jeroen Staudt.

**Writing – review & editing:** Jeroen Staudt, Yvonne Rensen, Hein De Haan, Boukje Dijkstra, Jos Egger.

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
