## [Decision Letter · Decision Letter 0]

28 May 2025

PONE-D-25-06883No relation between Body Mass Index and neurocognitive recovery in abstinent alcohol dependent patients?PLOS ONE

Dear Dr. Staudt,

Thank you for submitting your manuscript to PLOS ONE. After careful consideration, we feel that it has merit but does not fully meet PLOS ONE’s publication criteria as it currently stands. Therefore, we invite you to submit a revised version of the manuscript that addresses the points raised during the review process.  Please submit your revised manuscript by Jul 12 2025 11:59PM. If you will need more time than this to complete your revisions, please reply to this message or contact the journal office at plosone@plos.org . Please include the following items when submitting your revised manuscript:

We look forward to receiving your revised manuscript.

Kind regards,

Assoc. Prof. Phakkharawat Sittiprapaporn, Ph.D.

Academic Editor

PLOS ONE

Journal Requirements:

2. In the online submission form, you indicated that [The data that support the findings of this study contain potentially identifying and/or sensitive patient information and hence are only available upon reasonable request to the scientific committee of Tactus addiction treatment (weco@tactus.nl). The provision of data will be considered by the Tactus Local Scientific Research Committee and the authors involved in this study. The sharing of patient data is subject to Dutch and European legal and ethical regulations.].

Reviewers' comments:

Reviewer's Responses to Questions

**Comments to the Author**

1. Is the manuscript technically sound, and do the data support the conclusions?

Reviewer #1: Yes

Reviewer #2: Yes

2. Has the statistical analysis been performed appropriately and rigorously? 

Reviewer #1: Yes

Reviewer #2: Yes

3. Have the authors made all data underlying the findings in their manuscript fully available?

Reviewer #1: Yes

Reviewer #2: Yes

4. Is the manuscript presented in an intelligible fashion and written in standard English?

Reviewer #1: Yes

Reviewer #2: Yes

5. Review Comments to the Author

Reviewer #1: The manuscript" No relation between Body Mass Index and neurocognitive recovery in abstinent alcohol dependent patients?" was clearly written, uses validated cognitive measures, and addresses an important clinical question. However, the paper suffers by examining a factor, BMI, that is not a compelling predictor of cognitive function for several reasons. Other more precise and plausible measures of nutrition or other predictors of cognitive function in AUD were not explored due to practicality, but practicality alone is not sufficient justification when validity is in question.

BMI, is essentially a measure of your weight to height. One would assume that nutritional deficiency only occurs when BMI is low, as a high BMI indicates the person is consuming more food, and thus is getting more nutrients. While both low and high BMI are unhealthy, they are unhealth for different reasons. In the paper you combine these groups together for sample size. This manipulation obscures important nutritional states and lumps them into one unhealthy category that may obscure effects present in the high and low BMI groups. I recommend either examining the model with low, normal, and high as separate categories and accept that low BMI has low sample size, or remove the low BMI group altogether in a sensitivity analysis, and only interpret reduced subject model. In addition, you mention that there is a possible u-shape relationship between BMI and cognition, but do not attempt to model it this way. Why not?

Line 248-250 – “Just over half of the patients (35 / 61) performed equal or better on the second MoCA screening.” This is the same as saying there was no observed difference. Please update this.

Table 2: The N’s listed are less than 86. There was no discussion on how missing data was handled or why participants dropped out. Please include more information of participant flow through the trial and how missing data was handled.

Table 3: No SDs after mean score in unhealthy column.

Results section: “Association between BMI and neurocognitive functioning”. You found an effects of the TMT5 card, however when you added years of AUD to the model, this was no longer significant. This implies that years of AUD predicted TMT5 do some degree. Discuss the relationship between years of AUD, BMI, and Cognition.

On this note, you had the opportunity to check if any of the demographic information, such as age, years of AUD, etc, were predictors of cognition on their own, yet did not explore this.

Sample and power: The sample size is appropriate for a single model, however 12 models were run which will inflate type 1 error. Personally, I think given the exploratory nature and general non-results of this study, this is fine, however, it needs to be stated this way. Either correct for multiple comparisons or state more explicitly this is exploratory and corrections for multiplicity were not performed. Consider Helm of FDR techniques if you decide to correct. Overall, this should not really affect the results of this paper as there were few significant findings anyway.

Line 214: (Enter method, p < 0.5). is a typo.

Overall, the main issue with this study is that is essentially is reporting a non-result. It concludes that BMI is not a good prediction of cognitive function in AUD. While there was given justification in the introduction as to why this is a plausible relationship, the study centers on the idea that AUD affects nutrition which affects cognition. BMI is a poor measure of nutrition as you can be nutritionally poor as a low BMI or fine at a high BMI, while being unhealthy at both ends. This is a complex interaction which is not fully explored in this paper.

Revisions should in general bolster the importance of the result that BMI is not a good predictor of cognition in AUD. In addition to this, you need to establish that the way you measured analyzed BMI supports this conclusion.

Reviewer #2: Thank you for the opportunity to review this manuscript. This study explores the possibility of BMI as a predicter of cognitive function in AUD patient. The manuscript was well written. Unfortunately, there is concern regarding the conflation of nutritional deficiency and nutritional status. While the paper has merit and addresses an interesting topic, it would benefit from revisions below:

- Introduction

o From my understandings of this section, the authors chose BMI as the representation of nutritional status, to seek the relationship between nutritional deficiencies and cognitive function. The authors also noted about the increased BMI in people with excessive alcohol use. Since BMI can range from healthy to unhealthy status, they as a whole are not the equivalent of ‘nutritional deficiency.’ Hence, the statement regarding what BMI could represent is a little confusing. Later on in the manuscript, I’m not sure if the authors intended to use healthy BMI as control, but I still think that the revision is needed. If the authors want BMI to be the representation of nutritional status, both good and bad, there should be more background literature about the previous evidence relating to this point (BMI and nutritional status in general).

o Please add the evidence of how the use of tobacco should be included in this study. Can tobacco cause neurocognitive function decline as well? There are many other substances that patients with AUD could have used. Why was only tobacco chosen?

- Materials and methods:

o Procedure

The authors should provide more background information about the inpatient alcohol detoxification program such as duration of the program.

From my understandings that there were total of 4 tests (two for MoCA, d2 Test of Attention, and D-KEFS trail making test), the authors should state clearly when and how each tests was conducted. From “During the sixth week patients underwent extensive neuropsychological assessment (Line 146)," my understanding is that all of the four tests were conducted in the 6th week. It would be easier to understand if the authors could explain “extensive neuropsychological assessment” more in details.

o Measurements

Montreal Cognitive Assessment (MoCA)

• “Standardly two different versions were used during the two assessment points” (Line 172): This sentence was unclear. Please explain which versions, why and how they were used. Was a participant asked the same version for both assessment points?

• Please explain more about the scoring system of the test. What is the maximum score? Does the higher score mean cognitive impairment or the other way around?

• Please give information on the cut-off of cognitive impairment. Did the participants have cognitive impairment at the beginning of the program?

D2 Test of Attention and D-KEFS trail making test

• Please give more information as mentioned above

• Please explain why only MoCA was conducted at two time points, but not all the test.

o Statistical analyses

Since the idea would be ‘if a participant undergoes detoxification program (=abstinence), the cognitive function should improve and BMI might be able to predict that,’ I think adding the variables related to alcohol level would give more insight on the state of abstinence. (As the authors mentioned urinalysis on line 148, I reckon there is data that can be used.)

Line 213, please clarify what is MoCA total score. Are they the sum of the 1st assessment and the 2nd one?

Please explain why BMI at a single time point was used instead of the difference of BMI before and after the program (= changes in nutritional status.)

For the first research question, if a participant undergoes the same test (or not?) at two time points, should mixed model regression be used instead?

I think that if the authors could add the changes in BMI to the analyses, it would give more insights on the relationship between BMI and neurocognition function. (The hypothesis would be that if a patient has become abstinent, their nutritional status and neurocognition function should improve altogether.)

- Discussion

o Regarding the statement on Line 345 about current and past nutritional status, the use of change of BMI at the beginning and the assessment point might be able to give more information.

o As already stated in the section, more sample size, especially the patients with low BMI could have given more insights on the topic.

6. PLOS authors have the option to publish the peer review history of their article (what does this mean? ). If published, this will include your full peer review and any attached files.

**Do you want your identity to be public for this peer review?** For information about this choice, including consent withdrawal, please see our Privacy Policy .

Reviewer #1: No

Reviewer #2: No

---

## [Author Response · Author response to Decision Letter 1]

23 Jul 2025

POINT TO POINT REPLY

Reviewer 1

1. Reviewer #1: The manuscript" No relation between Body Mass Index and neurocognitive recovery in abstinent alcohol dependent patients?" was clearly written, uses validated cognitive measures, and addresses an important clinical question. However, the paper suffers by examining a factor, BMI, that is not a compelling predictor of cognitive function for several reasons. Other more precise and plausible measures of nutrition or other predictors of cognitive function in AUD were not explored due to practicality, but practicality alone is not sufficient justification when validity is in question. BMI, is essentially a measure of your weight to height. One would assume that nutritional deficiency only occurs when BMI is low, as a high BMI indicates the person is consuming more food, and thus is getting more nutrients. While both low and high BMI are unhealthy, they are unhealthy for different reasons. In the paper you combine these groups together for sample size. This manipulation obscures important nutritional states and lumps them into one unhealthy category that may obscure effects present in the high and low BMI groups. I recommend either examining the model with low, normal, and high as separate categories and accept that low BMI has low sample size, or remove the low BMI group altogether in a sensitivity analysis, and only interpret reduced subject model. In addition, you mention that there is a possible u-shape relationship between BMI and cognition, but do not attempt to model it this way. Why not?

Authors reply: We thank the reviewer for their comment. We agree that combining the groups with very low and very high BMI might obscure results that might become visible when examining these groups individually. We adopted the suggestion of the reviewer, choose different qualifications and adjusted our analysis. In the revised manuscript we now examine three categories of BMI (very low, normal and very high) separately. The text has been revised accordingly, resulting in removing all the terms of ‘healthy’ or ‘unhealthy’ from the manuscript and changing the statistical analysis, the results and part of the discussion. The most important changes are the following:

Introduction section:

“Given the possibility of an inverted U-shaped relation we also examine whether categories of BMI (very low, normal, very high) predict neurocognitive function.” (lines 130 – 131)

Method section, subsection Statistical analysis:

“In order to examine the relation between BMI (as a continuous or a categorical measurement) and neurocognitive functioning , we used multiple linear regression analysis (Enter method, p < 0.05). BMI categories were examined using dummy variables, with the normal BMI category as the reference group.” (lines 255 – 260)

We replaced Table 3 with a new version, see appendix.

Results section:

“A regression model predicting performance on TMT card 2 by means of categories of BMI yielded a significant result (R2 = 0.064, F (2, 70) = 2.397, p = 0.098). The contribution of the very high BMI category was significant (β = 0.235, p = 0.054), the very low BMI category did not contribute significantly (β = -0.050, p = 0.673). No other determinants could be identified as a candidate confounder in this relationship.

A regression model predicting performance on TMT card 5 by means of categories of BMI yielded a significant result (R2 = 0.065, F (2, 70) = 2.427, p = 0.096). The contribution of the very high BMI category was significant (β = 0.217, p = 0.074), the very low BMI category did not contribute significantly (β = -0.087, p = 0.468).

Two candidate confounders were identified. Adding age to the equation yielded a significant model (R2 = 0.106, F (3, 69) = 2.738, p = 0.050), with a positive effect of age on the dependent variable (β = 0.206, p = 0.078). The beta coefficient of the very low BMI category decreased > 10% from -0.087 to -0.075. Adding Age of Onset Of Daily Alcohol Use to the model did not yield a significant model (R2 = 0.085, F (3, 53) = 1.631, p = 0.193).” (see lines 356 – 370)

“Results from linear mixed model analysis yielded no significant effect of time (p = .452), BMI category (p = .689) or time * BMI category (p = .168) on repeated measures of both MoCA assessments.” (see lines 385 – 387)

Discussion section:

“With regard to our first research question, we found that both the continuous, as the categorical BMI value, correlated with a task mainly tapping speed and visuomotor skills (TMT-5). The categorical BMI value also correlated with a comparable task tapping digit sequencing, next to speed and visuomotor skills (TMT-2). Visuospatial and motor impairments may endure after detoxification [25,48,49] and may be related to BMI at admission. Interestingly, the results showed differences in the contribution of the categories of BMI (very low, normal and very high) to the models. The category of very low BMI was negatively related to performance on both the TMT-2 and TMT-5 tasks, the category of a very high BMI on the other hand was positively related to the outcomes. Although only the contribution of the very high BMI group was significant, possibly due to the very small sample size of the very low BMI group (N = 9), these results may be indicative of a difference in performance between the groups with the very low group performing worse. This may partly support an inverted U-shaped relation between BMI and speed -and visuomotor performance. The results furthermore show that the factor Age should be considered as a confounder in this relationship. With increasing age, performance on one task (TMT-5) increased. Since older adults show an increased chance of visual -and motor control deficits [50], this finding seems counterintuitive and might be attributed to the aforementioned skewed BMI data.” (lines 399 – 415)

“With regard to our second research question, the results did not support evidence for a relation between BMI at admission with recovery of neurocognitive function, as measured with two cognitive screening instruments during short (2 weeks) and mid-term (6 weeks) abstinence.” (lines 431 – 434)

2. Line 248-250 – “Just over half of the patients (35 / 61) performed equal or better on the second MoCA screening.” This is the same as saying there was no observed difference. Please update this.

Authors reply: We understand this notion and have removed these lines.

3. Table 2: The N’s listed are less than 86. There was no discussion on how missing data was handled or why participants dropped out. Please include more information of participant flow through the trial and how missing data was handled.

Authors reply: We agree that this point should be further elaborated in the manuscript. We updated the nature of our study and data handling. We described the character of this study, the inclusion criteria, the missing data on demographic variables and missing data on our outcome measures.

“retrospective cohort” (line 141)

“Participants who met criteria for a DSM-5 AUD disorder [28] were included for analyses”. (lines 151 – 152)

“Since missing variables were missing at random, these were listwise removed from analyses.” (lines 259 – 260)

“A total of 86 patients were included in this study. Data on almost all demographic variables were available, with exception of ‘years of daily alcohol use’ (N = 59), ‘age of onset of daily alcohol use’ (N = 63) and ‘neurological injury’ (N = 77). Due to variations in the administered tests during the seven years of data collection, 23 participants did not complete the first MoCA, 26 did not complete the second MoCA, 26 participants did not complete the d2, 13 participants did not complete the TMT, and 14 participants did not complete the TMT card 4.” (lines 292 – 298)

4. Table 3: No SDs after mean score in unhealthy column.

Authors reply: We thank the reviewer for pointing this out. The standard deviations are now added to the new Table 3 (added as an attachment given the landscape format).

5. Results section: “Association between BMI and neurocognitive functioning”. You found an effects of the TMT5 card, however when you added years of AUD to the model, this was no longer significant. This implies that years of AUD predicted TMT5 do some degree. Discuss the relationship between years of AUD, BMI, and Cognition.

Authors reply: We agree with the reviewer that years of AUD may have predicted TMT-5 to a certain degree and was indeed considered as a candidate confounder. However, since it eventually did not cause the beta value to change > 10% in the model, it is not considered as a confounder. Given the total amount of candidate confounders (years of AUD, Age and Age of onset of daily alcohol use) found in the different models (TMT-2 and TMT-5), and in order to be concise we chose to only elaborate on Age as this factor is considered as a confounding factor.

6. On this note, you had the opportunity to check if any of the demographic information, such as age, years of AUD, etc, were predictors of cognition on their own, yet did not explore this.

Authors reply: The variables mentioned may indeed be predictors of one of the outcome measures, however these were only analyzed for their relation with our dependent variables when they also were related with BMI. This was part of the analysis to select candidate confounders, as described in our analysis section. In order to keep focus, we therefore did not primarily address other demographic predictors and their relation with our outcome measures (see lines 244 – 254).

7. Sample and power: The sample size is appropriate for a single model, however 12 models were run which will inflate type 1 error. Personally, I think given the exploratory nature and general non-results of this study, this is fine, however, it needs to be stated this way. Either correct for multiple comparisons or state more explicitly this is exploratory and corrections for multiplicity were not performed. Consider Helm of FDR techniques if you decide to correct. Overall, this should not really affect the results of this paper as there were few significant findings anyway.

Authors reply: We agree and we more explicitly mentioned the explorative nature of the study in the introduction, discussion and conclusion section.

“The aim of this explorative study is to examine the relation between BMI and neurocognitive functioning in patients with AUD.” (lines 127 – 128)

“Given the explorative nature of this study, no correction of type I error was carried out.” (line 287)

“Although for all other analysis our sample size, used to perform our statistical analysis, was similar to the sample size used in other relevant studies [11–14], we ran multiple (12) models, risking type I error. Therefore the (limited) positive results cannot be excluded as false positives and need replication.” (see lines 514 – 518)

8. Line 214: (Enter method, p < 0.5). is a typo.

Authors reply: We think the reviewer points at the inconsistency found regarding the level of significance, which should be “Enter method, p < 0.15)”. This is now adjusted (line 232)

9. Overall, the main issue with this study is that is essentially is reporting a non-result. It concludes that BMI is not a good prediction of cognitive function in AUD. While there was given justification in the introduction as to why this is a plausible relationship, the study centers on the idea that AUD affects nutrition which affects cognition. BMI is a poor measure of nutrition as you can be nutritionally poor as a low BMI or fine at a high BMI, while being unhealthy at both ends. This is a complex interaction which is not fully explored in this paper.

Revisions should in general bolster the importance of the result that BMI is not a good predictor of cognition in AUD. In addition to this, you need to establish that the way you measured analyzed BMI supports this conclusion.

Authors reply: We very much see the point and have addressed this issue as mentioned in question 7, and by adding the following lines in the discussion:

“All in all, in the current study, we generally found no evidence for a relation between BMI and neurocognitive function, or recovery of neurocognitive functions, after six weeks of abstinence.” (lines 443 – 445)

“Although BMI is considered an appropriate measure reflecting nutritional deficiencies, BMI alone is not considered a valid predictor of neurocognition in patients with AUD for several reasons.” (see lines 461 – 462)

“ To conclude, this study found generally no support for a relation between BMI at admission and neurocognitive function, or recovery of neurocognitive functions, after six weeks of abstinence.” (lines 520 – 522)

Reviewer 2

Thank you for the opportunity to review this manuscript. This study explores the possibility of BMI as a predicter of cognitive function in AUD patient. The manuscript was well written. Unfortunately, there is concern regarding the conflation of nutritional deficiency and nutritional status. While the paper has merit and addresses an interesting topic, it would benefit from revisions below:

- Introduction

1. From my understandings of this section, the authors chose BMI as the representation of nutritional status, to seek the relationship between nutritional deficiencies and cognitive function. The authors also noted about the increased BMI in people with excessive alcohol use. Since BMI can range from healthy to unhealthy status, they as a whole are not the equivalent of ‘nutritional deficiency.’ Hence, the statement regarding what BMI could represent is a little confusing. Later on in the manuscript, I’m not sure if the authors intended to use healthy BMI as control, but I still think that the revision is needed. If the authors want BMI to be the representation of nutritional status, both good and bad, there should be more background literature about the previous evidence relating to this point (BMI and nutritional status in general).

Authors reply: We recognize the importance of explaining whether BMI reflects nutritional status adequately. We expanded the existing explanation by describing how good health can be distinguished from poor health (i.e. malnutrition), specifically by discussing the linear and U-shaped relationship between BMI, malnutrition and cognitive functioning:

“BMI has been significantly related to nutritional status (by means of the Subjective Global Assessment) in patients with [18] and without AUD [19,20], is associated with alcohol related complications such as liver disease [1], and is part of standard physical examination in addiction healthcare (according to the guidelines from the National Institute for Health and Care Excellence [NICE], Alcohol-use disorders: diagnosis and management) [21,22]. Yet, only two studies investigated the relation between BMI and neurocognitive functioning, by measuring BMI at the start of a clinical admission and assessments of neurocognitive functioning (MoCA) during their stay [11,12]. In addition, the relation between BMI and neurocognitive functioning may not be linear, since excessive alcohol use seems to increase BMI to a certain point, after which alcohol consumption accelerates metabolism and decreases fat mass, causing an inverse relation with BMI [17]. Several mechanisms may underlie this reversed U-shape, e.g. an increased microsomal detoxification or reduced lipogenesis (see [17]). Though an exact tipping point is unknown, this curve is also being studied in other areas of healthcare [23] and may be used to model the relation between malnutrition and neurocognitive functioning.” (lines 86 – 101)

2. Please add the evidence of how the use of tobacco should be included in this study. Can tobacco cause neurocognitive function decline as well? There are many other substances that patients with AUD could have used. Why was only tobacco chosen?

Authors reply: we thank the reviewer for pointing us to substantiating the use of confounding factors such as tobacco smoking. Since we include a number of other potential confounding factors in our study (e.g. gender, education, age of onset, etc.), we decided to refer to previously written systematic reviews that elaborate on these factors and their relation

---

## [Decision Letter · Decision Letter 1]

22 Aug 2025

PONE-D-25-06883R1No relation between Body Mass Index and neurocognitive recovery in abstinent alcohol dependent patients?PLOS ONE

Dear Dr. Staudt,

Thank you for submitting your manuscript to PLOS ONE. After careful consideration, we feel that it has merit but does not fully meet PLOS ONE’s publication criteria as it currently stands. Therefore, we invite you to submit a revised version of the manuscript that addresses the points raised during the review process.

We look forward to receiving your revised manuscript.

Kind regards,

Assoc. Prof. Phakkharawat Sittiprapaporn, Ph.D.

Academic Editor

PLOS ONE

**Journal Requirements:**

Reviewers' comments:

Reviewer's Responses to Questions

**Comments to the Author**

1. If the authors have adequately addressed your comments raised in a previous round of review and you feel that this manuscript is now acceptable for publication, you may indicate that here to bypass the “Comments to the Author” section, enter your conflict of interest statement in the “Confidential to Editor” section, and submit your "Accept" recommendation.

Reviewer #1: All comments have been addressed

Reviewer #2: All comments have been addressed

2. Is the manuscript technically sound, and do the data support the conclusions?

Reviewer #1: Yes

Reviewer #2: Yes

3. Has the statistical analysis been performed appropriately and rigorously? 

Reviewer #1: Yes

Reviewer #2: Yes

4. Have the authors made all data underlying the findings in their manuscript fully available?

Reviewer #1: Yes

Reviewer #2: Yes

5. Is the manuscript presented in an intelligible fashion and written in standard English?

Reviewer #1: Yes

Reviewer #2: Yes

6. Review Comments to the Author

**Reviewer #1:**  The revised manuscript has adequately addressed the concerns raised by me and the other reviewers. This study was well done and provides an important addition to the area of AUD recovery. I recommend this article to be published. In my final pass however, I did find a few small issues to be addressed. I do not need to review again after these changes are made. The line number below refer to he red-lined version.

Line - 209 "Normality was judged by visual inspection of the data. A skewness and kurtosis between -2 and +2 were accepted for normality." The fact that you calculated a number to compare to +/-2 suggestess more was done than visual inspection.

Line - 244 "All significant relations (p < 0.15)", You are using a non-standard alpha criterion. Add in the text a justification for not using 0.05.

Line- 400 "as the categorical BMI value", should be "and the categorical BMI value"

Line - 443 " all-in-all", replace or remove this phrase

**Reviewer #2:**  (No Response)

7. PLOS authors have the option to publish the peer review history of their article (what does this mean? ). If published, this will include your full peer review and any attached files.

**Do you want your identity to be public for this peer review?** For information about this choice, including consent withdrawal, please see our Privacy Policy .

Reviewer #1: No

Reviewer #2: No

---

## [Author Response · Author response to Decision Letter 2]

29 Aug 2025

Reviewer 1

1. "Line - 209 "Normality was judged by visual inspection of the data. A skewness and kurtosis between -2 and +2 were accepted for normality." The fact that you calculated a number to compare to +/-2 suggestess more was done than visual inspection.

Authors reply: we clarified this by adding the following "In case of doubt, skewness and kurtosis were calculated to Z-scores (see for example [47]), where a score between -2 and +2 was accepted for normality [48]" (lines 227 – 229)

2. Line - 244 "All significant relations (p < 0.15)", You are using a non-standard alpha criterion. Add in the text a justification for not using 0.05.

Authors reply: we adjusted our explanation in line 238 to clarify this value. “A more tolerant p-value (p < 0.15) is used to minimalize type II error. “ (line 238)

3. Line- 400 "as the categorical BMI value", should be "and the categorical BMI value"

Authors reply: we have changed this as suggested.

4. Line - 443 " all-in-all", replace or remove this phrase"

Authors reply: we have changed this as suggested.

Reviewer 2

No comments

---

## [Decision Letter · Decision Letter 2]

18 Sep 2025

No relation between Body Mass Index and neurocognitive recovery in abstinent alcohol dependent patients?

PONE-D-25-06883R2

Dear Dr. Staudt,

We’re pleased to inform you that your manuscript has been judged scientifically suitable for publication and will be formally accepted for publication once it meets all outstanding technical requirements.

Kind regards,

Assoc. Prof. Phakkharawat Sittiprapaporn, Ph.D.

Academic Editor

PLOS ONE

Additional Editor Comments (optional):

Reviewer #1:

Reviewer #2:

Reviewers' comments:

Reviewer's Responses to Questions

**Comments to the Author**

1. If the authors have adequately addressed your comments raised in a previous round of review and you feel that this manuscript is now acceptable for publication, you may indicate that here to bypass the “Comments to the Author” section, enter your conflict of interest statement in the “Confidential to Editor” section, and submit your "Accept" recommendation.

Reviewer #1: All comments have been addressed

Reviewer #2: All comments have been addressed

2. Is the manuscript technically sound, and do the data support the conclusions?

Reviewer #1: Yes

Reviewer #2: Yes

3. Has the statistical analysis been performed appropriately and rigorously? 

Reviewer #1: Yes

Reviewer #2: Yes

4. Have the authors made all data underlying the findings in their manuscript fully available?

Reviewer #1: Yes

Reviewer #2: Yes

5. Is the manuscript presented in an intelligible fashion and written in standard English?

Reviewer #1: Yes

Reviewer #2: Yes

6. Review Comments to the Author

Reviewer #1: 1. All comments addressed. 2. The manuscript was technically sound. 3. The statistics were performed rigorously. 4. The authors have made the data available upon request. 5. The manuscript is well written. No additional comments, I approved this manuscript for publication on an earlier revision and this version of the manuscript has not changed significantly to invalidate my earlier recommendation.

Reviewer #2: (No Response)

7. PLOS authors have the option to publish the peer review history of their article (what does this mean? ). If published, this will include your full peer review and any attached files.

**Do you want your identity to be public for this peer review?** For information about this choice, including consent withdrawal, please see our Privacy Policy .

Reviewer #1: No

Reviewer #2: No

---

## [Editor Report · Acceptance letter]

PONE-D-25-06883R2

PLOS ONE

Dear Dr. Staudt,

I'm pleased to inform you that your manuscript has been deemed suitable for publication in PLOS ONE. Congratulations! Your manuscript is now being handed over to our production team.

Kind regards,

on behalf of

Assoc. Prof. Dr. Phakkharawat Sittiprapaporn

Academic Editor

PLOS ONE